# OZITX, a pertussis toxin-like protein for occluding inhibitory G protein signalling including Gα$_z$

Alastair C. Keen [1,2,3,13], Maria Hauge Pedersen[4,5,6,13], Laura Lemel[2,3], Daniel J. Scott [7,8], Meritxell Canals[2,3], Dene R. Littler[9], Travis Beddoe [10], Yuki Ono[11], Lei Shi[12], Asuka Inoue [11], Jonathan A. Javitch [4,5,14✉] & J. Robert Lane [2,3,14✉]

Heterotrimeric G proteins are the main signalling effectors for G protein-coupled receptors. Understanding the distinct functions of different G proteins is key to understanding how their signalling modulates physiological responses. Pertussis toxin, a bacterial AB$_5$ toxin, inhibits Gα$_{i/o}$ G proteins and has proven useful for interrogating inhibitory G protein signalling. Pertussis toxin, however, does not inhibit one member of the inhibitory G protein family, Gα$_z$. The role of Gα$_z$ signalling has been neglected largely due to a lack of inhibitors. Recently, the identification of another Pertussis-like AB$_5$ toxin was described. Here we show that this toxin, that we call OZITX, specifically inhibits Gα$_{i/o}$ and Gα$_z$ G proteins and that expression of the catalytic S1 subunit is sufficient for this inhibition. We identify mutations that render Gα subunits insensitive to the toxin that, in combination with the toxin, can be used to interrogate the signalling of each inhibitory Gα G protein.

[1] Drug Discovery Biology, Monash Institute of Pharmaceutical Sciences, Monash University, Parkville, VIC 3052, Australia. [2] Division of Physiology, Pharmacology and Neuroscience, School of Life Sciences, Queen's Medical Centre, University of Nottingham, Nottingham, UK. [3] Centre of Membrane Proteins and Receptors, University of Birmingham and University of Nottingham, Nottingham, UK. [4] Departments of Psychiatry and Molecular Pharmacology and Therapeutics, Vagelos College of Physicians and Surgeons, Columbia University, New York, NY, USA. [5] Division of Molecular Therapeutics, New York State Psychiatric Institute, New York, NY, USA. [6] NNF Center for Basic Metabolic Research, Section for Metabolic Receptology, Faculty of Health and Medical Sciences, University of Copenhagen, Copenhagen, Denmark. [7] Department of Biochemistry and Molecular Biology, University of Melbourne, Parkville, VIC 3052, Australia. [8] The Florey Institute of Neuroscience and Mental Health, University of Melbourne, Parkville3052VIC 3052, Australia. [9] Infection and Immunity Program and Department of Biochemistry and Molecular Biology, Biomedicine Discovery Institute, Monash University, Clayton, VIC 3052, Australia. [10] Department of Animal, Plant and Soil Science and Centre for AgriBioscience, La Trobe University, Bundoora, VIC 3086, Australia. [11] Graduate School of Pharmaceutical Sciences, Tohoku University, Sendai, Miyagi 980-8578, Japan. [12] Computational Chemistry and Molecular Biophysics Section, National Institute on Drug Abuse - Intramural Research Program, National Institutes of Health, Baltimore, MD, USA. [13] These authors contributed equally: Alastair C. Keen, Maria Hauge Pedersen. [14] These authors jointly supervised this work: Jonathan A. Javitch, J. Robert Lane. ✉email: Jonathan.Javitch@nyspi.columbia.edu; rob.lane@nottingham.ac.uk

Heterotrimeric guanine nucleotide-binding proteins (G proteins) are important signalling transducers that link cell-surface receptors such as G protein-coupled receptors (GPCRs) to intracellular effectors[1–3]. They consist of a Gα subunit as well as Gβ and Gγ subunits that function as an obligate dimer. There are four subfamilies of Gα subunits (Gα$_s$, Gα$_i$, Gα$_q$ and Gα$_{12}$) based on sequence similarity. Their functions can be broadly generalised based on this classification. The stimulatory (Gα$_s$) and the inhibitory (Gα$_i$) subfamilies stimulate and inhibit adenylate cyclases, respectively[1,4]. The Gα$_q$ subfamily activates phospholipase C-β leading to increases in cytosolic Ca$^{2+}$, and the Gα$_{12}$ subfamily activates Rho family GTPases that regulate cytoskeletal processes[2,5]. Understanding the distinct signalling roles of individual members of each subfamily is central to our comprehension of how they control different physiological processes.

The Gα subunit, and in particular its carboxy tail, is largely responsible for determining the specificity of the interaction with an activated GPCR[6,7]. The GPCR acts as a guanine nucleotide exchange factor, promoting the exchange of bound GDP for GTP at the guanine nucleotide-binding domain of the Gα subunit. This causes the Gα subunit to dissociate from, or rearrange relative to, the Gβγ dimer, and both then act on downstream effectors[8,9]. The Gα subunit is a GTPase, hydrolysing GTP to restore GDP to the binding domain, allowing the Gβγ dimer to reassociate and completing the cycle.

AB$_5$-type toxins have proved to be useful tools for the interrogation of G protein signalling. These toxins are characterised by a hetero-hexameric structure consisting of the enzymatically active A subunit and pentameric ring of B subunits, which are responsible for recognition of host cell-surface receptors and facilitate cellular entry. In order to modulate host cell behaviour, AB$_5$ toxins have varied actions on their targets, including protease activity[10], RNA N-glycosidase activity[11] and ADP-ribosylation[12]. Of relevance to G-protein signalling, cholera toxin acts on the Gα$_s$ subfamily[13] and Pasteurella multocida *Pasteurella multocida* toxin acts on the Gα$_i$, Gα$_q$ and Gα$_{12}$ family to render them constitutively active[14]. Pertussis toxin (PTX), from *Bordetella pertussis*, ADP ribosylates all members of the Gα$_i$ subfamily, except for Gα$_z$[15]. Researchers have exploited these actions to identify the Gα subunits responsible for particular cell signalling processes. PTX-driven ADP ribosylation occurs on a cysteine residue of four residues from the carboxy terminus of Gα$_i$ subunits, rendering them incapable of coupling to GPCRs. PTX-insensitive Gα$_{i/o}$ subunit mutants, in which the cysteine modified by PTX is replaced by another residue, have been used to understand the role of individual Gα$_i$-subfamily members in vitro. One inhibitory G-protein family member, Gα$_z$, lacks this cysteine and is thus insensitive to PTX. Gα$_z$ has a slow GDP-GTP exchange rate, slow GTP hydrolysis rate, and a restricted pattern of expression[16–19]. Despite these unique characteristics, relatively little is known about the physiological role of Gα$_z$ signalling, although evidence has been provided for its roles in circadian behaviours, dopaminergic signalling, and pancreatic islet β cell biology[19–22]. Its function in other physiological processes remains elusive, in part due to its insensitivity to PTX. Indeed, there may be cases in which inhibitory G protein signalling has been ruled out based on a lack of PTX effect while neglecting the potential role of Gα$_z$.

A recent publication reported the identification and structural characterisation of a PTX-like protein derived from a uropathogenic *Escherichia coli*[23]. The toxin has an active A subunit homologous to that of PTX and has a similar overall structural fold (Supplementary Fig. 1). Application of this toxin to HEK 293 cells, African green monkey kidney cells or bovine brain lysate revealed its substrates as heterotrimeric G proteins[23]. Using Gα$_{i2}$

as a substrate in vitro, the toxin was shown to have distinct site(s) of ADP ribosylation from that of PTX—an asparagine residue and a lysine residue eight and ten residues from the carboxy terminus, respectively[23]. The asparagine is conserved across several Gα subunits, suggesting that the toxin may have broader substrate specificity than PTX.

In this study, we show that this toxin inhibits the coupling of all Gα$_{i/o/z}$ G proteins, including Gα$_z$. Thus, we refer to it as Gα$_O$, Gα$_Z$ and Gα$_i$ inhibiting ToXin, or in short; OZITX. The active A subunit is functional when expressed in mammalian cells, bypassing the need for toxin purification. Moreover, we generate mutants of the members of the Gα$_i$ subfamily that are OZITX insensitive, and hence, can serve as tools in combination with OZTIX treatment to investigate the role of individual Gα$_{i/o/z}$ G proteins.

## Results

### OZITX treatment abolishes GPCR-mediated activation of all Gα$_i$ subfamily members, including Gα$_z$.

We hypothesised that OZITX may display a broader selectivity as compared to PTX because the Asn$^{348}$ residue that is ADP ribosylated by OZITX is conserved in a greater number of Gα subunits as opposed to the cysteine modified by PTX, which is only present in Gα$_i$ and Gα$_o$ family members (Fig. 1a, b)[23]. We first sought to determine whether OZITX inhibits coupling to members of the inhibitory G protein subfamily. To achieve this, we used a previously described bioluminescence resonance energy transfer (BRET) assay that measures the release of Gβγ subunits from the Gα subunits upon activation of the heterotrimer (Fig. 2a)[24,25]. While this assay provides a method for rapidly assessing G protein activation, the signal may be partially contaminated by endogenously expressed Gα subunits[25,26]. We, therefore, adapted the assay for use in HEK293A CRISPR/Cas ΔGα-all cells in which all the Gα subunits had been genetically knocked out[27]. This allowed us to monitor the Gβγ release specifically from the activation of one Gα subtype of interest that had been exogenously transfected.

The dopamine D$_2$ receptor (D$_2$R) promiscuously couples to Gα$_{i/o}$ and Gα$_z$ G proteins[28,29]. Cells transiently expressing the D$_2$R were pre-incubated with PTX or OZITX followed by stimulation with the D$_2$-like receptor-selective agonist ropinirole[30]. We observed that OZITX completely blocked the activation of Gα$_{i1}$, Gα$_{i2}$, Gα$_{i3}$, Gα$_{oA}$ and Gα$_{oB}$ (Fig. 2b). Further, as predicted from the carboxy-tail Asn$^{348}$ residue presented in Gα$_z$, Gα$_z$ could no longer couple to the D$_2$R following OZITX treatment as well (Fig. 2b). In contrast, Gα$_z$ was insensitive to inhibition by pre-treatment of cells with PTX, consistent with the absence of the required cysteine residue (Fig. 1a). This finding extends the initial characterisation of OZITX, showing that unlike PTX it can inhibit Gα$_z$ as well as Gα$_{i/o}$[23].

Next, analogous experiments were performed with another Gα$_{i/o/z}$-coupled GPCR; the μ opioid receptor. HEK293A CRISPR/Cas ΔGα cells transiently expressing the MOPR were pre-incubated with either OZITX or PTX and then stimulated with the agonist DAMGO (Fig. 2c). OZITX inhibited coupling to each of the Gα$_{i/o/z}$ subunits completely (Fig. 2c). This showed, as expected, that OZITX does not discriminate between GPCRs when inhibiting Gα$_{i/o/z}$ G-protein activation.

We then sought to further characterise the toxin by measuring the activation of Gα$_{i2}$ by the D$_2$R after exposure to OZITX at different timepoints. Gα$_{i2}$ activation decreased with increasing time of OZITX exposure until Gα$_{i2}$ activation was completely abolished approximately sixteen hours after the addition of OZITX (Fig. 2d). This is consistent with the characteristics of PTX and suggests that OZITX, like PTX, would be best utilised in the laboratory by incubating with the cells for more than 16 h (Supplementary Fig. 2).

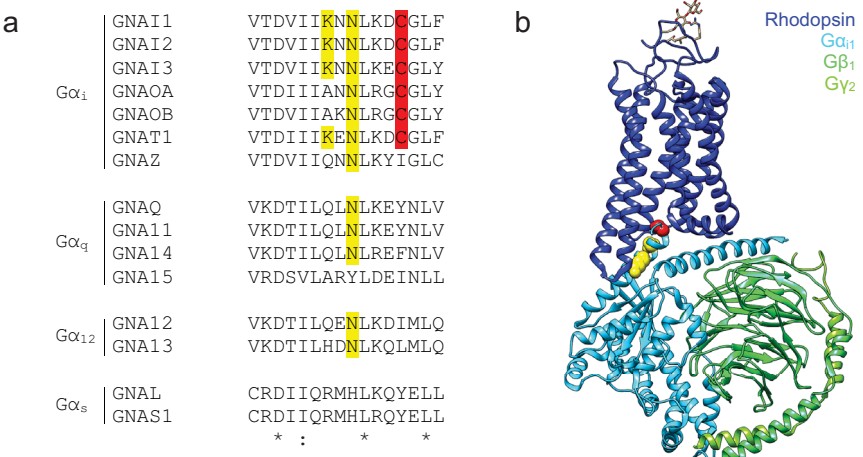

**Fig. 1 Identification of Gα carboxy-tail amino acid residues that are putatively ADP-ribosylated by OZITX. a** Amino acid sequence alignment of carboxy-termini residues of heterotrimeric Gα proteins. Sequences were aligned with Clustal Omega version 1.2.4. '*' represents a completely conserved residue. ':' represents a conserved residue (>0.5 in the Gonnet PAM 250 matrix). '.' represents a weakly conserved residue (≤0.5 and >0 in the Gonnet PAM 250 matrix). Cysteine residues ADP ribosylated by PTX are indicated in red. Putative lysine and asparagine residues ADP ribosylated by OZITX identified by Littler and colleagues[23] are indicated in yellow. The asparagine residue that is a putative substrate is conserved across many Gα subunits. **b** The location of OZTX's and PTX's substrate amino acid sites within a GPCR-G protein complex. The structure of rhodopsin bound to $Gα_{i1}β_1γ_2$ is depicted in the cartoon (PDB code 6CMO). Rhodopsin is shown in dark blue, $Gα_{i1}$ in light blue, $Gβ_1$ in green and $Gγ_2$ in light green. The carboxy-terminal $Cys^{351}$ residue ADP ribosylated by PTX is shown in red spheres. $Lys^{345}$ and $Asn^{347}$, the putative residues ADP ribosylated by OZITX, are highlighted in yellow spheres. Graphic constructed using UCSF chimera.

**OZITX does not ablate $Gα_s$, $Gα_q$ or $Gα_{12}$ subfamily coupling.**
In addition to the inhibitory Gα G protein subfamily, the asparagine eight residues from the carboxy terminus is conserved in other Gα subfamily members (Fig. 1a). We therefore sought to further assess the substrate selectivity of OZITX across all Gα subunits. The $Gα_s$ subfamily possesses a histidine residue instead of an asparagine in this position. In accordance with OZITX's proposed mechanism of action, overnight incubation with OZITX did not inhibit $Gα_s$ or $Gα_{olf}$ activation by the dopamine $D_1$ receptor, a $Gα_{s/olf}$-coupled receptor, stimulated with the $D_1R$-selective agonist SKF83822 (Fig. 3a)[31–34].

$Gα_q$, $Gα_{11}$ and $Gα_{14}$, but not $Gα_{15}$, possess an asparagine residue eight residues from their C termini, so one might expect these three subunits to be substrates for OZITX (Fig. 1a). We measured the activation of the $Gα_q$ subfamily proteins by the $Gα_q$-coupled neurotensin receptor 1 stimulated by the agonist NT8-13 with and without OZITX treatment[35,36]. OZITX pre-treatment did not inhibit the activation of $Gα_q$, $Gα_{11}$ or $Gα_{15}$ although $Gα_{14}$ activation was slightly (~25%) decreased (vehicle control = 0.0840, OZITX-treated = 0.0644, $P = 0.0012$, one-way ANOVA with Dunnett's multiple comparisons test) (Fig. 3b).

Both members of the $Gα_{12}$ subfamily; $Gα_{12}$ and $Gα_{13}$, also have asparagine as their eighth to last residue (Fig. 1a). The neurotensin receptor 1 is known to couple to the $Gα_{12}$ subfamily[37]. While we were successful in detecting robust activation of $Gα_{12}$ and $Gα_{13}$, there was no inhibitory effect on the activation of either subunit when the cells were treated with OZITX (Fig. 3c). Taken together, we conclude that despite the presence of this asparagine residue at the C-terminus of $Gα_q$ and $Gα_{12}$ subfamily members, no detectable inhibitory action of OZITX was observed, with the exception of the limited inhibition of $Gα_{14}$.

**Inhibition of cAMP production by $Gα_{i2}$-, $Gα_{oA}$- and $Gα_z$ is inhibited by OZITX.** Cell-surface receptor signalling is commonly amplified in subsequent steps down the signalling cascade. We wanted to confirm that the apparent complete blockade of $Gα_{i/o/z}$ signalling at the level of G-protein coupling would concord with measurements further downstream. We assessed the effect of OZITX treatment in measurements of intracellular cAMP levels using an intramolecular conformational BRET sensor of cAMP (CAMYEL), since the $Gα_i$ subfamily bind and inhibit adenylate cyclases[4,38]. In these experiments, we used HEK293A cells that harbour a genetic knockout of all the $Gα_i$ subfamily members using CRISPR/Cas (HEK293A CRISPR/Cas $ΔGα_i$)[27]. Individual $Gα_i$ subunits of interest were then transfected into this cell background. Cells were treated with forskolin to stimulate adenylate cyclase, followed by treatment with ropinirole to stimulate the $D_2R$, leading to activation of the $Gα_{i/o/z}$ subunit of interest. In the absence of a transfected Gα subunit, there was no detectable drug-induced inhibition of cAMP production (Fig. 4a). When $Gα_{i2}$ or $Gα_{oA}$ were transfected, activation of the $D_2R$ produced a decrease in relative cAMP levels (indicated by an increase in BRET ratio) and this was completely abolished in cells treated with OZITX (Fig. 4b, c). Cells transfected with $Gα_z$ also produced a decrease in cAMP, albeit to a slightly smaller degree, and this was again blocked in the presence of OZITX (Fig. 4d). This confirms that OZITX-mediated ADP ribosylation inhibits downstream $Gα_{i/o/z}$-mediated signalling.

**The active A subunit of OZITX can be transfected into mammalian cells to act as an inhibitor.** In order to treat cells with $AB_5$ toxin protein complexes, both expression and purification of this toxin are required[23]. The active A subunit of PTX alone can be transiently expressed to inhibit $Gα_{i/o}$ signalling[39,40]. Accordingly, we tested whether the OZITX would be functional upon transfection of the cDNA encoding the active A subunit alone (OZITX-S1), thus increasing its accessibility and utility to laboratories. The cDNA sequence of OZITX-S1 was codon-optimised for high expression in human cells and co-transfected into HEK293T cells along with the $D_2R$, the WT $Gα_{i/o/z}$ subunits and the G-protein activation sensors. Upon activation of the $D_2R$ with the agonist quinpirole the responses in cells transfected with $Gα_{i1-3}$, $Gα_{oA}$ and $Gα_{oB}$ were inhibited in cells transfected with the positive control PTX-S1 cDNA as well as the OZITX-S1 cDNA (Fig. 5a–c and Supplementary Fig. 3). Importantly, while transfection of cells with OZITX-S1 inhibited $Gα_z$ activation,

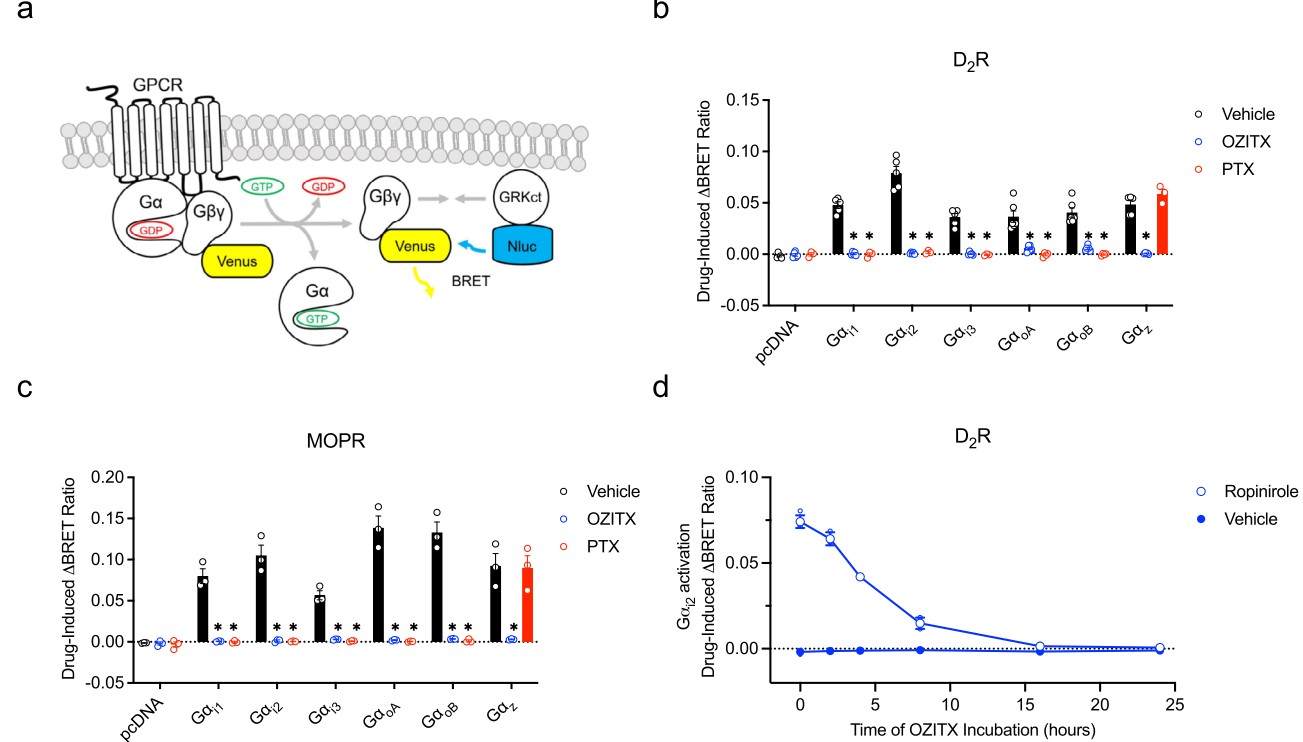

**Fig. 2 Activation of members of the Gαi subfamily in the presence of OZITX and PTX. a** Representation of the BRET sensors used for detection of G-protein activation. The Gαβγ heterotrimer is activated through agonist binding to the GPCR and the Gα and Gβγ-venus dissociate. Free Gβγ-venus is bound by masGRK3ct-Nluc that serves as a BRET donor resulting in non-radiative energy transfer from Nluc to venus. **b** Dopamine D$_2$receptor (D$_2$R)-mediated activation or (**c**) μ opioid receptor (MOPR)-mediated activation of Gα$_i$ subfamily members in the presence of OZITX or PTX. HEK293A CRISPR/Cas ΔGα-all cells expressing the D$_2$R or MOPR were pre-treated with either vehicle (black), OZITX (blue) or PTX (red) for 24 h. Cells were stimulated with 1 μM ropinirole (D$_2$R) or 1 μM DAMGO (MOPR) for 2.5 min followed by BRET detection. Data represent the mean drug-induced increase in BRET ratio from vehicle ± SEM from six independent experiments (D$_2$R) or three independent experiments (MOPR). *Represents where the response is significantly different ($P < 0.001$) from the respective vehicle toxin untreated control condition (black bar) as determined by a one-way ANOVA with Dunnett's multiple comparisons test. **d** Time course of OZITX treatment on G-protein activation. HEK 293 ΔGα-all cells were transfected with cDNA encoding the D$_2$R, Gα$_{i2}$ and G-protein activation sensors. Cells were pre-treated with OZITX for the indicated times. BRET was measured 2.5 min after stimulation with 1 μM ropinirole (blue open circles) or vehicle (blue filled circles). The basal BRET ratio prior to agonist stimulation has been subtracted to give the drug-induced ΔBRET ratio. Data represent the mean ± SD from three separate experiments. Individual replicates are shown as small circles.

transfection of PTX-S1 cDNA did not. Having shown that transfected OZITX-S1 is functional, we then confirmed the pattern of OZITX selectivity across the Gα subfamilies was in accord with our previous results using treatment with the complete OZITX protein complex (Supplementary Fig. 3). Consistent with these findings, the OZITX-S1 transfection was ineffective in abolishing the activation of Gα$_s$, Gα$_q$ and Gα$_{12}$ subfamilies (Supplementary Fig. 3). In contrast to the partial inhibition of Gα$_{14}$ that we observed when using the purified OZITX (Fig. 3b), we did not observe inhibition of Gα$_{14}$ in experiments transiently expressing the OZITX-S1 (Supplementary Fig. 3).

ADP ribosylation of the C-terminal cysteine of Gα$_{i/o}$ subunits by PTX is thought to inhibit the functional interaction between these G proteins and an activated GPCR. To test whether ADP ribosylation might also inhibit such an interaction we measured the recruitment of Gα$_{oA}$:G$_β$:G$_γ$-venus G protein heterotrimers to the D$_{2L}$R in the presence of either PTX-S1 or OZITX-S1 expression. Our data clearly show that both PTX and OZITX inhibit Gα$_{oA}$:G$_β$:G$_γ$-venus G-protein heterotrimer recruitment to the D$_2$R but that only OZITX inhibits Gα$_z$:G$_β$:G$_γ$-venus G-protein heterotrimer recruitment, as expected. This is consistent with a mechanism of action whereby ADP ribosylation of the C-terminal asparagine of Gα$_{i/o/z}$ by OZITX prevents their coupling to GPCRs akin to the action of PTX at the C-terminal cysteine of Gα$_{i/o}$ (Supplementary Fig. 4).

**Gα$_i$ subunits can be made OZITX insensitive for dissection of Gα$_{i/o/z}$ subtype signalling specificity.** Understanding the actions of a single Gα$_{i/o/z}$ subtype can be challenging because there are usually multiple Gα$_{i/o/z}$ members expressed within any given cell type. A method that has permitted the investigation of the role of individual Gα$_{i/o/z}$ subunits in a particular signalling process, such as coupling to a specific GPCR, is the use of PTX-insensitive Gα$_i$ mutants in combination with PTX to uncouple any endogenously expressed PTX-sensitive Gα$_i$ subunits[41]. We, therefore, wanted to generate OZITX-insensitive Gα$_{i/o/z}$ mutants in the hope of increasing the scope of OZITX applications. To render the Gα$_{i/o/z}$ subunits insensitive to OZITX, we replaced the asparagine eight residues from the carboxy terminus to an alanine (Gα$_{i1}$-N347A, Gα$_{i2}$-N348A, Gα$_{i3}$-N347A, Gα$_{oA}$-N347A, Gα$_{oB}$-N347A and Gα$_z$-N348A) as this residue was previously identified as the most likely substrate site (Fig. 1a)[23]. We then performed G protein-activation assays using the D$_2$R to activate each Gα$_i$ mutant in the presence or absence of PTX-S1 or OZITX-S1 (Fig. 5 and Supplementary Figs. 5 and 6). In contrast to the activation of the wild-type Gα$_{i3}$, Gα$_{oA}$ and Gα$_z$ that are all abolished by OZITX (Fig. 5a–c), activation of Gα$_{i3}$-N347A, Gα$_{oA}$-N347A and Gα$_z$-N348A were OZITX insensitive (Fig. 5d–f). When these mutant Gα subunits were transfected the potency of quinpirole was similar to that observed in the case of the WT Gα subunit, suggesting that these mutations did not affect the efficiency with which they couple to

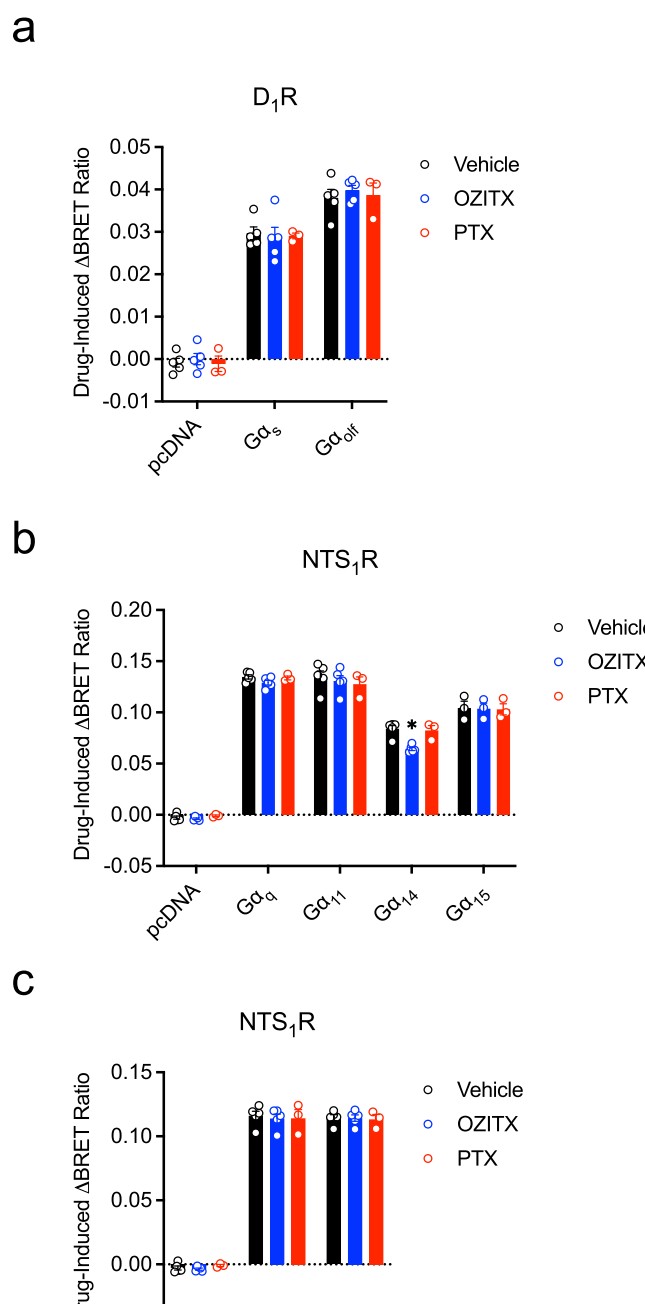

**Fig. 3 Gαs, Gαq and Gα12 subfamily activation in presence of OZITX and PTX. a** Activation of Gαs subfamily members by the dopamine D1 receptor (D1R) in the presence of OZITX and PTX. **b** Activation of Gαq subfamily members by NTS1R in the presence of OZITX and PTX. **c** Activation of Gα12 subfamily members by the neurotensin receptor 1 (NTS1R) in the presence of OZITX and PTX. HEK 293 ΔGα-all CRISPR cells were transfected with cDNA encoding the particular GPCR and Gα together with the G-protein activation sensors as described in "Methods". The cells were pre-treated with either vehicle (black), OZITX (blue) or PTX (red) for 24 h before stimulation with the agonists 100 nM SKF83822 (D1R)/1 μM NT8-13 (NTS1R) for 2.5 min followed by BRET detection. The data are represented as the mean drug-induced increase in BRET ratio from vehicle control ± SEM from three separate experiments. *Represents where the OZITX or PTX-treated condition is significantly different (P < 0.001) from the vehicle-treated condition (black) as determined by a one-way ANOVA with Dunnett's multiple comparisons test.

the D2R. In addition, it was observed that the N347A/N348A mutation did not impact the PTX sensitivity of the Gαi subunits (Fig. 5d–f). Likewise, the well-characterised PTX-insensitive mutation (C351I) introduced into Gαi3 and GαoA, did not disturb the ability of OZITX to act on them (Fig. 5g, h). Having identified that the N347A/N348A mutation renders these Gαi members insensitive to OZITX without perturbation, the mutations were extended into the remaining Gαi subunits and validated (Supplementary Figs. 5 and 6).

**The C-terminal ten residues of Gαi are sufficient to confer OZITX selectivity.** The asparagine residue important for the inhibitory action of OZITX is present in both the Gi/o/z and the Gq subfamilies (with the exception of Gα15). Thus, the selective action of OZITX at the Gi/o/z G-protein α subunits appears not to be determined solely by the presence or absence of this residue. Similarly, a previous study has shown that replacement of the 5 C-terminal residues of Gαq with that of Gαi (which includes the cysteine that is modified by PTX) allows this chimeric Gαqi5 G protein to couple to Gi protein family-coupled receptors but, importantly, does not confer sensitivity to PTX[42]. This suggests that there are determinants of PTX in addition to the presence of this cysteine residue. To explore other determinants of OZITX selectivity we generated two chimeric G proteins in which the last 10 (Gαqi10) or 13 (Gαqi13) residues of Gαq were replaced with those of Gαi3, a region that includes the asparagine residue modified by OZITX. In an assay measuring Ca²⁺ mobilisation, OZITX was unable to inhibit the Ca²⁺ response of the muscarinic M3 acetylcholine receptor co-expressed with Gαq when activated by the agonist carbachol (Supplementary Fig. 7). In this Ca²⁺ mobilisation assay we were unable to detect a measurable response to the agonist ropinirole when the D2R was co-expressed with Gαq, but we observed responses when the D2R was co-expressed with both chimeric G proteins, meaning the D2R can couple to both Gαqi10 and Gαqi13. Interestingly, both PTX and OZITX could inhibit these responses, indicating that the last 10 residues of Gαi are sufficient to confer the sensitivity to both PTX and OZITX inhibition on a Gαq background (Supplementary Fig. 7). While both PTX and OZITX partially inhibited the Gαqi10 Ca²⁺ signal, the expression of both toxins completely inhibited Gαqi13 signalling. This pattern is consistent with the idea that the greater the amount of C-terminal Gαi amino acids swapped with those of Gαq, the better the chimeric Gαqi proteins become as substrates for both αβ5 toxins. However, the lower potency of ropinirole in the Gαqi13 Ca²⁺ assay suggests that the coupling of the D2R to this chimera is less efficient, which may also account for the apparently greater inhibitory effect of PTX and OZITX (Supplementary Fig. 7).

## Discussion

For decades, PTX and CTX have proven to be useful tools in GPCR signalling research to interrogate the Gα protein subfamilies or even specific Gα proteins responsible for particular physiological processes. Here, we characterise and demonstrate the utility of OZITX, a recently identified AB5 toxin, for the inhibition of GPCR-mediated activation of the Gαi/o/z subfamily. Importantly, unlike PTX, this activity extends to include Gαz. OZITX acts to ADP-ribosylate an asparagine in the C-terminus of Gαi/o/z proteins, a site distinct from the cysteine modified by PTX, accounting for this broader specificity. We found that OZITX displays a selective action to completely inhibit Gαi/o/z proteins with no activity at Gαs, Gαq or Gα12 proteins, with the exception of limited inhibition of Gα14. The catalytic subunit of PTX (PTX-S1) can be expressed in mammalian cells to effectively inhibit Gαi/o signalling, avoiding the time and cost associated with acquiring the purified protein[39,40]. We demonstrate that the catalytic OZITX-S1 subunit can be used in a similar

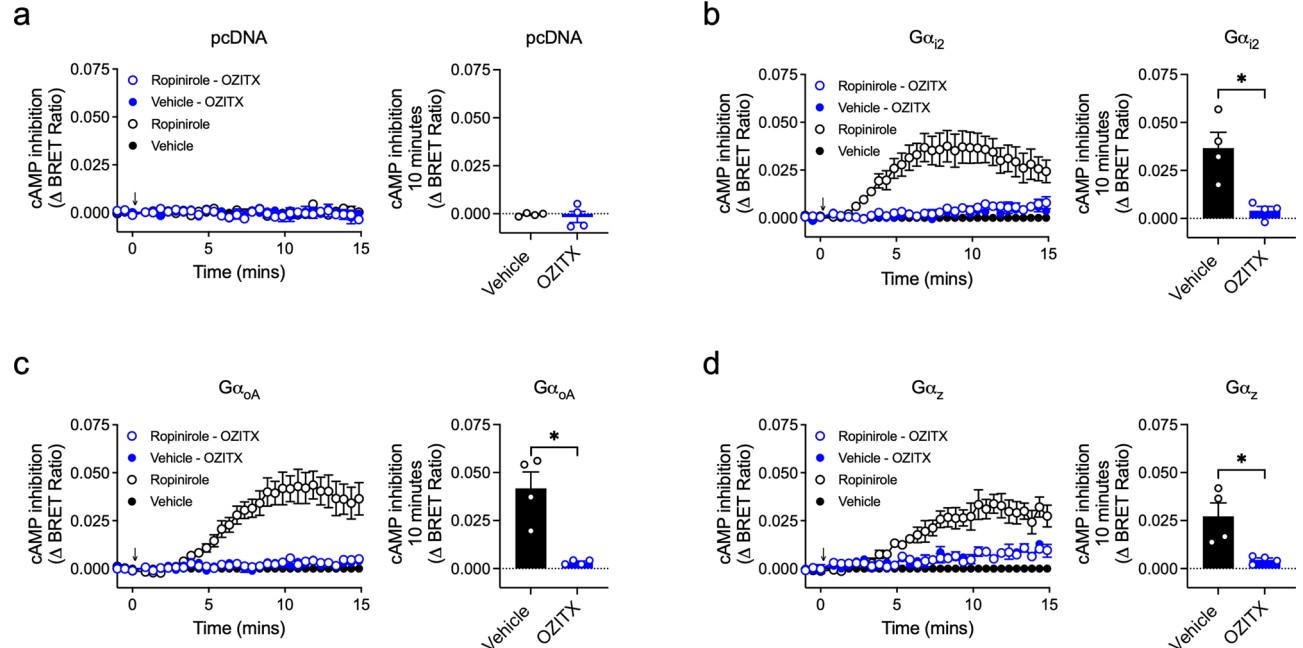

**Fig. 4 The effect of OZITX on Gαi2-, GαoA- and Gαz-mediated inhibition of cAMP production.** Inhibition of forskolin-stimulated cAMP production was detected in live cells using CAMYEL; a conformational BRET sensor based on EPAC. HEK 293 ΔGα$_{i/o}$ CRISPR cells were transfected with DNA encoding the D$_2$R, CAMYEL and either (**a**) pcDNA3.1+ control, (**b**) Gα$_{i2}$, (**c**) Gα$_{oA}$ or (**d**) Gα$_z$. Transfected cells were then incubated with either vehicle (black) or OZITX (blue) for 24 h. Cells were then pre-stimulated with 10 μM forskolin for 10 min before stimulation with either vehicle control (filled circles) or 1 μM ropinirole (open circles). Data are baseline-corrected to the cells not treated with OZITX or ropinirole and is shown as the mean ± SEM from four separate experiments. Measurements of cAMP inhibition between vehicle and OZITX-treated conditions were compared using an unpaired Student's $t$ test * represents statistical significance $P < 0.05$ (pcDNA –$P = 0.700$, Gα$_{i2}$ –$P = 0.008$, Gα$_{oA}$ – $P = 0.004$, Gα$_z$ – 0.019).

manner, increasing the utility of this tool. We identify mutations within Gα$_{i/o/z}$ subfamily members that render them insensitive to OZITX and maintain their ability to couple to GPCRs. Together, these tools can be used to identify the Gα$_{i/o/z}$ subunits participating in defined signalling pathways.

PTX played an important role in identifying the Gα$_i$ subfamily by distinguishing it from the Gα$_s$ subfamily[4]. PTX was shown to block the inhibitory effect that Gα$_i$ proteins have on adenylyl cyclases, thus building evidence for a separate Gα species with distinct functionality to Gα$_s$. Since then, PTX has been widely used with the same rationale, that is, to differentiate GPCR responses mediated by Gα$_i$ proteins from other signal transducers[43]. However, such an approach cannot exclude the possibility that Gα$_z$ might contribute to a particular response since it is insensitive to PTX[17,18]. A clear advantage of OZITX, then, is that it can inhibit Gα$_z$ in addition to inhibiting Gα$_{i1}$, Gα$_{i2}$, Gα$_{i3}$ and the Gα$_o$ isoforms. We have not evaluated whether OZITX inhibits the coupling of the visual and taste Gα subunits; Gα$_{t1}$, Gα$_{t2}$ and Gα$_{gust}$. One might expect ADP ribosylation by OZITX to occur on Gα$_{t1}$, Gα$_{t2}$ and Gα$_{gust}$ since they harbour an asparagine as their eighth to last amino acid residue in addition to having high sequence homology to the other Gα$_i$ subunits, although given our findings that not all Gα subunits that contain this asparagine are inhibited by OZITX, inhibitor activity at Gα$_{t1}$, Gα$_{t2}$ and Gα$_{gust}$ must be determined experimentally.

Our findings suggest that OZITX could serve as a replacement for PTX in most experimental paradigms aimed at interrogating Gα$_{i/o/z}$ G-protein signalling. There are, however, cases where PTX and OZITX can be used in parallel due to their different Gα specificities, for example when disentangling the functions of Gα$_z$ from the other Gα$_{i/o}$ subunits. OZITX-treated, PTX-treated and -untreated conditions run in parallel would enable the signalling mediated by Gα$_z$, PTX-sensitive Gα$_i$ subunits and toxin-insensitive Gα subunits to be isolated.

Previous studies aimed at interrogating Gα$_z$ signalling have relied on other strategies, including overexpression of Gα$_z$-specific RGS proteins[19], Gα$_z$-directed siRNA[44], and Gα$_z$ de-activation via PKC phosphorylation[45]. However, unlike OZITX, these approaches do not completely block activation of Gα$_z$ so the influence of residual Gα$_z$ signalling cannot be excluded, particularly when looking at an effect further down an amplified signalling cascade. Genetic knockouts of the gene that encodes Gα$_z$ have been used for this reason but are technically challenging as compared to OZITX treatment[20,46]. In addition, the results of such knockout approaches may be confounded by adaptive changes to the cell and/or circuit over time that compensate for the loss of that particular protein. The advantage of OZITX is that it can be used in a relatively acute manner following overnight treatment, so its use is less likely to be confounded by compensatory changes in cell function.

The substrate site that is ADP ribosylated by OZITX was shown to be an asparagine eight residues from the C-terminus of the Gα subunit[23]. In agreement with this, we showed that Gα$_i$ subunits can be made OZITX insensitive through mutation of the aligned asparagine in this position. Within our set of experiments, we observed that these mutations did not affect the potency or magnitude of the measured response as compared to when the WT Gα was used. This indicates that this mutation has not changed the coupling efficiency between the receptor and Gα subunit. It should be acknowledged, however, that these observations may be both receptor and downstream effector dependent. It may be that other mutations at this position may be superior to the alanine mutation for a particular combination, as has been observed for analogous studies using PTX and PTX-insensitive Gα mutants[47]. Nonetheless, the OZITX-insensitive mutants can serve as a useful tool in combination with OZITX treatment to investigate the signalling of particular Gα$_{i/o/z}$ proteins in isolation. In our hands, mutation of the Asn$^{347/348}$

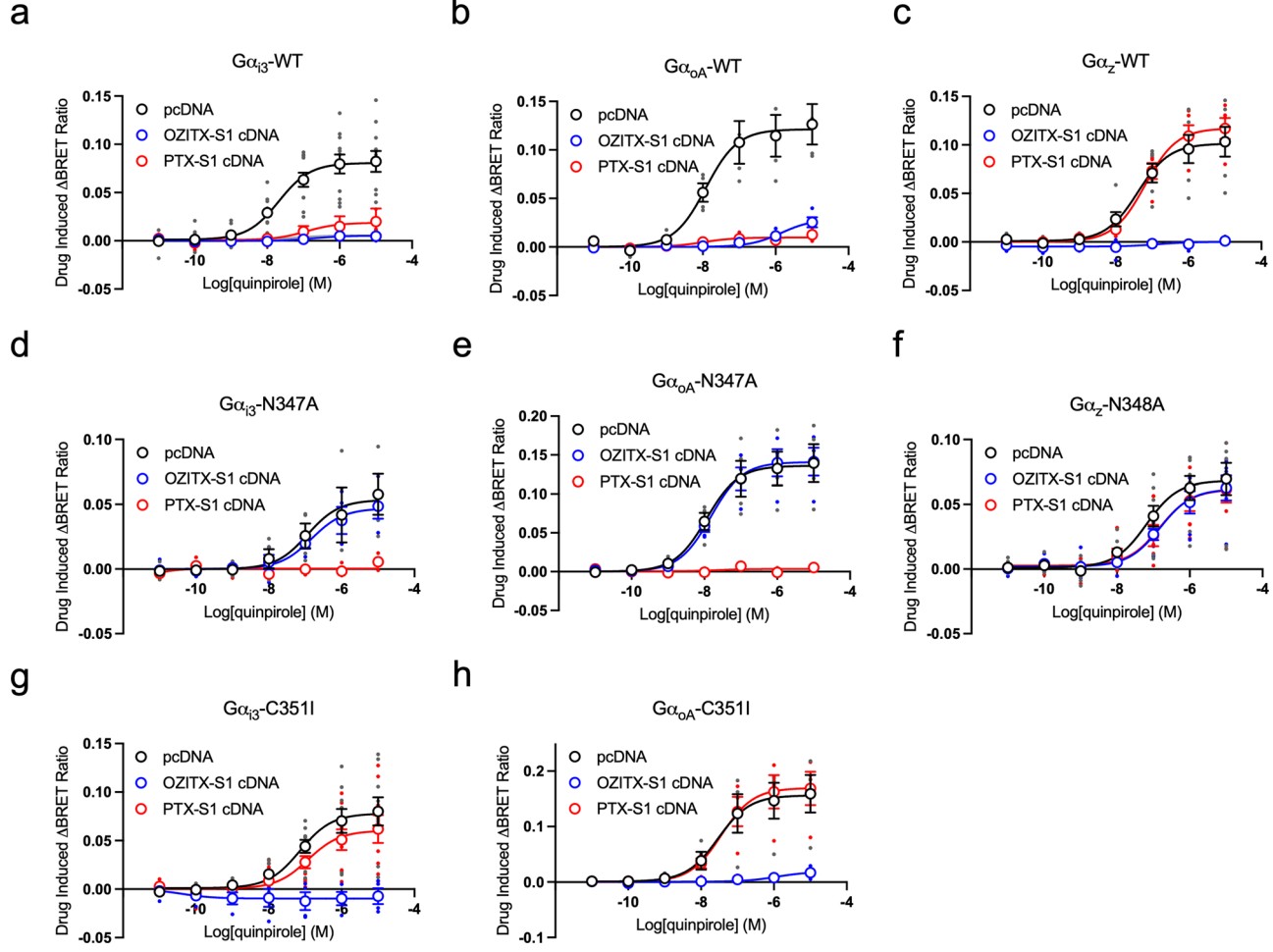

**Fig. 5 OZITX sensitivity of Gαi subfamily carboxy-tail Asn347/348 mutants.** **a** Gα$_{i3}$-WT activation, $n = 11$. **b** Gα$_{oA}$-WT activation, $n = 4$. **c** Gα$_z$-WT activation, $n = 5$. **d** Gα$_{i3}$-N347A activation, $n = 4$. **e** Gα$_{oA}$-N347A activation, $n = 4$. **f** Gα$_z$-N347A activation, $n = 6$. **g** Gα$_{i3}$-C351 activation, $n = 4$. **h** Gα$_{oA}$-C351 activation, $n = 4$. The G-protein activation assay was performed on WT, Asn347Ala/Asn348Ala (putative OZITX site) and Cys351Ile (PTX-insensitive) mutants. Cells were transfected with the D$_2$R, the particular Gα mutant, the G-protein activation sensors and either a pcDNA3.1+ control (black open circles), OZITX-S1 cDNA (blue open circles) or PTX-S1 cDNA (red open circles). Cells were then stimulated with increasing concentrations of quinpirole before BRET detection. Data represent the mean drug-induced increase in BRET ratio from vehicle ± SEM. Individual replicates are shown as small circles.

residue alone was sufficient to render Gα$_{i1}$, Gα$_{i2}$ and Gα$_{i3}$ resistant to OZITX. These Gα subunits contain a lysine residue as their tenth-to-last residue (Lys[345/346]) that was suggested to also be a site for OZITX-mediated ADP ribosylation by Littler and colleagues (Fig. 1a)[23]. Our results suggest that this Lys[345/346] site is either a secondary substrate site that is minimally ADP ribosylated by OZITX, that ADP ribosylation of this site has no effect on G-protein coupling despite this residue being in close proximity to the GPCR upon coupling, or that the ribosylation of this lysine occurs sequentially to that of Asn[347/348] (Fig. 1b), such that the mutation of the asparagine residue is sufficient to abrogate reaction.

We hypothesised that the presence of an asparagine residue eight residues from the C-terminus of various Gα subunits would confer sensitivity to OZITX in a similar manner to the way in which the presence of Cys[351/352] confers sensitivity to PTX. In agreement with this, Gα$_{i/o/z}$ proteins were inhibited by OZITX whereas Gα$_s$, which lacks this asparagine, was not. We observed, however, that OZITX only had a small inhibitory effect on Gα$_{14}$ activation and had no effect on the remaining Gα$_q$ and Gα$_{12}$ subunits despite the presence of the aligned asparagine. Thus, this asparagine is not the only determinant of selectivity. This lack of

OZITX sensitivity can be reconciled either with OZITX not ADP-ribosylating the asparagine residue in G$_q$ and G$_{12}$ family Gα subunits or with ADP ribosylation of this residue in Gα$_q$ and Gα$_{12}$ not affecting GPCR coupling and signalling. Prior studies have shown that swapping the five carboxy-terminal residues of Gα$_{i2}$ or Gα$_{oA}$ onto Gα$_q$ is not sufficient to confer sensitivity to PTX, even though the modified Gα$_q$ contains the required cysteine residue four amino acids from the carboxy-termini[42]. This indicates that carrying the required substrate amino acid site is not sufficient to render the Gα subunit sensitive to PTX-like AB$_5$ toxins. Our own experiments using chimeras in which the last ten or thirteen amino acids of Gα$_q$ were swapped with those of Gα$_{i2}$ revealed that exchange of this region was sufficient to confer both PTX and OZITX sensitivity. We computationally explored the feasibility of ADP-ribosylation of Gα subunits in the D$_2$R-Gα$_i$ and the 5HT$_{2A}$-Gα$_q$ complexes, by covalently docking the ADP-ribose moiety on the asparagine residue using the available cryo-EM structures of these complexes[48,49]. Our results reveal that the 5HT$_{2A}$-Gα$_q$ complex can easily accommodate the ADP-ribosylated asparagine whereas this covalent docking approach could not identify any feasible pose for the D$_2$R-Gα$_i$ complex without steric clash (Supplementary Fig. 8). Even though

ADP-ribosylation of the asparagine can be sterically accommodated in the 5HT$_{2A}$–G$\alpha_q$ complex, we cannot rule out the possibility that this might still impact coupling, and as we have noted above, the residue may simply not be ADP-ribosylated due to the absence of other key determinants beyond the asparagine itself. Further studies are required to understand the additional structural basis for the recognition of specific G$\alpha$ subunits by AB$_5$-type toxins such as OZITX and PTX and to understand the selective action of OZITx for G$_{i/o/z}$ family G proteins.

Our study illustrates the continuing value in the characterisation and use of AB$_5$ toxins as laboratory tools. Host–pathogen interactions are hotspots of molecular evolution that result in proteins with extraordinary functionality. This is exemplified in the diversity of actions of ADP-ribosylating AB$_5$ toxins including PTX and CTX and now OZITX and their resulting value as research tools.

## Methods

**Materials**. Polyethylenimine (PEI), Linear (MW 25,000) was purchased from Polysciences, Inc. Ropinirole was purchased from BetaPharma (Shanghai) Co. Ltd. DAMGO ((D-Ala$^2$, N-Me-Phe$^4$, Gly-ol$^5$)-enkephalin) was purchased from Mimotopes. SKF83822, neurotensin residues 8-13 (NT8-13), (−)-quinpirole hydrochloride (#1061), acetylcholine chloride (#A2661), carbachol (C4382), D-glucose (#G8270) and pertussis toxin (PTX) were purchased from Sigma-Aldrich. Isoproterenol (#1747) and endothelin-1 (#1160) were purchased from Tocris Bioscience (Bristol, UK). Coelenterazine-h was purchased from both NanoLight™ Technology and Dalton research molecules (#50303-86-9). Forskolin was purchased from Cayman Chemicals (#11018). Nano-Glo® luciferase assay system, containing the furimazine substrate, was purchased from Promega.

**Plasmids**. pcDNA3.1(+) encoding human constructs of long isoform of the dopamine D$_2$ receptor (D$_2$R), μ opioid receptor (MOPR), dopamine D$_1$ receptor (D$_1$R), neurotensin receptor 1 (NTS$_1$R), M$_1$ muscarinic acetylcholine receptor (M$_1$R), β$_2$-adrenergic receptor (β$_2$AR), endothelin A receptor (ET$_A$R), G$\alpha_{i1}$, G$\alpha_{i2}$, G$\alpha_{i3}$, G$\alpha_{oA}$, G$\alpha_{oB}$, G$\alpha_z$, G$\alpha_{sS}$, G$\alpha_{sL}$, G$\alpha_{olf}$, G$\alpha_q$, G$\alpha_{11}$, G$\alpha_{14}$, G$\alpha_{15}$-EE, G$\alpha_{12}$ and G$\alpha_{13}$ were from the cDNA Resource Centre (cDNA.org). pcDNA3L-His-CAMYEL was purchased from ATCC (ATCC MBA-277). masGRK3ct-Nluc, masGRK3ct-Rluc8, venus-1-155-G$\gamma_2$ and venus-156-239-G$\beta_1$ were from Nevin Lambert, Augusta University. pCAGGS-Ric8A and pCAGGS-Ric8B were from Asuka Inoue, Tohoku University. The active S1 subunit of OZITX (EcPltAB) was codon-optimised, synthesised and inserted into pcDNA3.1 + (see Supplementary Note 1 for sequence). OZITX-resistant mutations were made in G$\alpha_{i1}$, G$\alpha_{i2}$, G$\alpha_{i3}$, G$\alpha_{oA}$ G$\alpha_{oB}$ and G$\alpha_z$ using site-directed mutagenesis. Primer sequences that were used for the mutagenesis can be found in Supplementary Table 1.

OZITX-resistant mutations were made by changing the eighth to last amino acid to an alanine (indicated in red) by using site-directed mutagenesis with the reverse primers used to the right, the alanine mutation change is shown in red and restriction sites chosen in blue (XhoI) or green (XbaI). The constructs were inserted into pcDNA3.1+ with KpnI and XhoI or XbaI as indicated. The two chimeric proteins G$\alpha_{qi10}$ and G$\alpha_{qi13}$ were generated using the Q5 site-directed mutagenesis kit from NEB. Primer sequences used for the mutagenesis can be found in Supplementary Table 2. PCR products were then treated with the KLD enzyme mix (kinase, ligase and DpnI) provided with the kit and then transform into NEB Turbo E. coli competent cells.

**Cell culture**. HEK293T cells were purchased from ATCC (CRL-3216). HEK293A ΔGα-all CRISPR/Cas knockout cells and HEK293A ΔG$\alpha_{i/o}$ CRISPR/Cas knockout cells were generated as described before[27]. HEK293T cells, HEK293A ΔGα-all cells and HEK293A ΔG$\alpha_{i/o}$ cells were cultured in T175 flasks with Dulbecco's Modified Eagle Medium (DMEM) + GlutaMAX™-I (Gibco, Invitrogen, Paisley, UK) with 10% foetal bovine serum (Corning #35-010) and 1% penicillin/streptomycin (Corning #30-002). All Cells were grown in a humidified incubator in 5% CO$_2$ at 37 °C and sub-cultured at a ratio of 1/10-1/20.

**Transfection**. Briefly, cells were harvested from T175 flasks and plated into six-well Nunc™ tissue culture plates at a density of 500,000 cells per well. The following day, the media was removed and replaced with fresh media and transfected using PEI as the transfection reagent. The corresponding amounts of PEI and DNA were added to the buffer separately before mixing together, incubating for 20 minutes, and then adding dropwise on top of the cells in the fresh media.

For the G-protein-activation assays where the toxin was added exogenously: The HEK293A ΔGα-all CRISPR knockout cells were transfected using PEI in a ratio of 6:1 PEI:DNA (w/w) in PBS. The cells were transfected with a cDNA mixture consisting of: 0.143 μg GPCR, 0.286 μg Gα, 0.143 μg Gβ$_1$-venus, 0.143 μg Gγ$_2$-venus, 0.143 μg masGRK3ct-Nluc and 0.143 μg Ric8A or Ric8B or pcDNA3.1.

The chaperone Ric8A was transfected together with G$\alpha_{14}$ and G$\alpha_{15}$ and Ric8B was transfected with G$\alpha_{olf}$.

For the cAMP BRET assays, where the toxins were exogenously added: The HEK293A ΔG$\alpha_{i/o}$ CRISPR knockout cells were transfected using PEI in a ratio of 6:1 PEI:DNA (w/w) in PBS. The cells were transfected with a cDNA mixture consisting of: 0.143 μg D$_2$R, 0.286 μg G$\alpha_{i2}$/G$\alpha_{oA}$/G$\alpha_z$/pcDNA3.1 and 0.429 μg CAMYEL sensor.

Assays, where the active A subunits of the toxins were transiently transfected, had the following conditions: HEK293T cells were transfected using PEI in a ratio of 1.5 PEI:1 DNA (w/w) mixed in 150 mM NaCl For the G-protein activation assays the cells were transfected with 0.500 μg β$_1$, 0.500 μg Venus-γ$_2$ and 0.100 μg masGRKctRluc8 as well as 1 μg of the G protein of interest together with 0.375 μg of a receptor suited for the specific G-protein class and 0.375 μg of the helper proteins Ric8A for G$\alpha_{14}$ and G$\alpha_{15}$ and Ric8B for G$\alpha_{olf}$ and finally 0.200 μg of either the active subunit of PTX (PTX-S1), OZITX (OZITX-S1) or pcDNA3.1+ as a control. For the cAMP production inhibition assays the cells were co-transfected with 1 μg of the CAMYEL sensor (ATCC MBA-277). For the Ca$^{2+}$ assay, the cells were transfected with 0.3 μg of the receptor (M3R or D2LR), 0.3 μg of G$\alpha_{q/qi10/qi13}$ and 0.2 μg of either S1-PTX/S1-OZITX or pcDNA3.

**G-protein activation**. G-protein activation was measured using a BRET assay that monitors Gβγ release[24,25]. The HEK293A ΔGα-all cells were first transfected as described earlier and the following day the cells were harvested and transferred into white 96-well CulturPlates (PerkinElmer) in DMEM + 10% FBS. In the conditions where the cells were treated with OZITX or PTX, the cells were left to adhere before being treated in the 96-well plate 16–20 h before performing the assay. The G-protein activation assays were then performed ~24 h after plating out the transfected cells. The media in each well was aspirated, washed with Hank's balanced salt solution pH 7.4 (HBSS), replaced with HBSS and then kept at 37 °C for the remainder of the assay. Furimazine was added with a multi-stepper pipette 15 min before agonist addition and left to equilibrate. The agonist was then added, and cells were incubated in a LUMIstar Omega (BMG Labtech) plate reader. The BRET measurements were then taken 2.5 min after agonist addition. Simultaneous dual emission filters were used in the LUMIstar Omega for detection of the luciferase at 445–505 nm and venus at 505-565 nm, all measured at 37 °C. For G-protein activation assays where the toxin active A subunit cDNAs were transfected, the same protocol was followed with some exceptions: HEK293T cells were used instead of CRISPR/Cas gene-edited cells, DPBS + 5 mM glucose was used as the assay buffer, 96-well black-white isoplates were used, and the plate was detected five minutes after agonist stimulation in a PHERAstar FS (BMG Labtech). After acquiring the data, the ratio of the venus emission channel was then divided by the luciferase emission channel to determine the BRET ratio. The vehicle-subtracted raw BRET ratio (drug-induced increase in BRET) is plotted for the G-protein activation assay data.

**Gγ-mVenus recruitment assay**. For the D$_2$R-mediated Gγ-mVenus recruitment assay, HEK293T cells were seeded onto six-well plates and transfected with a 1:6 ratio of DNA:polyethylenimine with plasmids encoding D2R-nluc (50 ng), G$\alpha_{oA/z}$ WT (cDNA resource centre, Bloomsburg University, PA, 125 ng), human Gβ$_1$ (250 ng) and human Gγ2- mVenus (250 ng). Cells were harvested from six-well plates 24 h after transfection and plated into poly-D-lysine coated (Sigma-Aldrich) white-bottom 96-well optiplates (Wallac, PerkinElmer Life and Analytical Sciences) at a density of 50,000 cells per well. Twenty-four hours after cells were transferred to plates, media was aspirated, cells washed once with DPBS and 80 μL DPBS containing 5 mM glucose was added to each well. In all, 10 μL of coelenterazine was added to each well and the plate read on the PHERAstar FS (BMG Labtech) for 5 min, paused for the addition of 10 μL agonists and read again for 10 min. After acquiring the data, the ratio of the venus emission channel was then divided by the luciferase emission channel to determine the BRET ratio. The vehicle-subtracted raw BRET ratio (drug-induced increase in BRET) is plotted for the G-protein recruitment assay data.

**Gα$_{i/o/z}$-mediated inhibition of cAMP production**. The cAMP production inhibition assays' principle is based on the ability of a genetically encoded conformational BRET sensor to detect the relative concentrations of intracellular cAMP[50]. Initially, the transfected HEK293A ΔG$\alpha_{i/o}$ cells were harvested and transferred into white 96-well CulturPlates in DMEM + 10% FBS 24 h after the transient transfection. When the cells were treated with OZITX or PTX, this occurred in the 96-well plate after adherence and about 18 h before the assay. Next, the cAMP inhibition assays were performed the following day after plating out the transfected cells and toxin or control treatment. On the day of the assay, the plate media was aspirated, washed once with HBSS pH 7.4 and replenished with HBSS pH 7.4 and then held at 37 °C for the rest of the experiment. In total, 5 μM coelenterazine-h was added 15 min before agonist addition. 10 μM Forskolin was added 10 min before agonist addition and the readings were then continuously taken in the live cells. Bioluminescence was detected on a LUMIstar Omega set to 37 °C. Simultaneous dual emission filters were used for the BRET donor at 445–505 nm and the acceptor at 505-565 nm. The ratio of the acceptor channel was then divided by the donor channel to determine the BRET ratio. The data was then

baseline-corrected to the vehicle control wells over time. A slightly modified protocol was followed for the assays where the active subunit cDNAs of the toxins were transfected: HEK293T cells were used instead of the HEK293A $\Delta G\alpha_{i/o}$ cells, 96-well black-white isoplates were used, DPBS + 5 mM glucose was used as the assay buffer, a higher concentration of 30 μM forskolin was used and this was co-added with the coelenterazine-h ten minutes prior to the addition of the agonist. The plate was then detected 20 min after agonist addition in a PHERAstar FS.

**$Ca^{2+}$ mobilisation assays.** Cells were seeded in a clear bottom black 96-well plate coated with poly-D-lysine (50 μg/ml) at 100,000 cells per well. The following day, cells were washed with 100 μl of 1× HBSS assay buffer supplemented with 10 mM glucose, 4 mM probenic acid at pH 7.4 and brilliant black, and then loaded with 100 μl Fluo-4 AM (1 μM) (prepared in DMSO and pluronic acid) for 45 min at 37 °C (no $CO_2$). The release of $Ca^{2+}$ was measured using a Flexstation 3 (Molecular Devices; Sunnyvale, CA). Drug dilutions were prepared in assay buffer (without Fluo-4) at 6x required concentration (volume 20 μl in 100 μl in Flexstation protocol) and transferred to a loading plate (transparent flat-bottom 96-well plate). Fluorescence was detected for 3 min 30 s at 485 nm excitation and 525 nm emission. Relative fluorescence units were normalised to the fluorescence stimulated by ionomycin to account for differences in cell number and loading efficiency.

**Data analysis, statistics and reproducibility.** GraphPad Prism 8 was used for data analysis and performing statistical tests. Statistical analysis was carried out with a Student's $t$ test or one-way ANOVA followed by a post hoc test where appropriate. $P$ values <0.05 were considered statistically significant. Data sets were of at least $n = 3$ and the experimental $n$ number is stated for each data set in the corresponding figure legend. Figures depicting molecular structures were constructed using ICM-Browser (MolSoft LLC) and UCSF Chimera[51]. The covalent docking was carried out with the covDock module of Schrodinger suite (version 2021-1), assuming a SN2 nucleophilic substitution reaction, which results in the a-orientation of the attached ADP-ribose moiety on the asparagine[52].

**Reporting summary.** Further information on research design is available in the Nature Research Reporting Summary linked to this article.

## Data availability
All original data have been deposited with *Communications Biology* in Supplementary Data 1 and are also available from the corresponding authors upon reasonable request.

## Materials availability
The cDNA for the S1 subunit of OZITX as described in the manuscript will be made available on reasonable request.

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

## Acknowledgements
This work was supported by the National Health and Medical Research Council (NHMRC, Australia, Project Grant 1049564), a Schaefer Research Scholars Program Award (Columbia University, New York, USA) and the Biotechnology and Biological Sciences Research Council (BB/T013966/1) to J.R.L. and National Institute of Health (NIH) grant MH54137 to J.A.J.

## Author contributions
A.C.K., M.H.P., J.A.J. and J.R.L. designed experiments. A.C.K., M.H.P. and L.L. performed experiments. A.C.K., M.H.P., L.L., J.A.J. and J.R.L. analysed the data. A.C.K., M.H.P., L.L., L.S., J.A.J., M.C. and J.R.L. wrote the manuscript. D.R.L., Y.O., T.B., A.I. and D.J.S. provided reagents.

## Competing interests
The authors declare no competing interests.
