## [Peer Review File · Communications Biology]

Reviewers' comments:

Reviewer #1 (Remarks to the Author):

The purpose of this study is to investigate the inhibitory function of OZITX for Gi/o/z and develop experimental systems to study the signaling of Gi/o/z individually. Here are a few comments to improve the manuscript.

1. Throughout the manuscript, place a space between number and unit. For example, 10uM should be 10 uM in the Figure S2 legend.
2. Please provide pcDNA control (i.e. Galpha-non transfected control) for Figure S3 and Figure S4 as shown in Figure 2 or Figure 3.
3. Please discuss a more detailed potential mechanism of non-inhibitory effects of OZITX on Gq or G12. A docking simulation of Gi/o/z and Gq or G12 binding at the active site of OZITX would show the potential mechanism.
4. In p 8, lane 169, it should be Fig. 4B & C, not 3B & C.
5. In p 9, lane 171, it should be Fig. 4D, not 3D.

Reviewer #2 (Remarks to the Author):

The authors characterized in their paper a pertussis toxin-like protein they called OZITX that impairs activation of the G proteins from the Gi family, including Gz. Through a series of functional studies, they showed that this toxin specifically inhibited Gi/o/z activation both when preincubated or when transfected into the cells. Gz inhibition is a novel feature compared to the PTX toxin that was already used to inhibit Gi proteins. As such, this work brings a new tool that, associated to the existing ones, will undoubtedly help dissect how the Gz pathway is selectively involved in the response to Gi-coupled receptors. Taken together, this is a therefore an interesting addition to the panel of tools used in GPCR signaling assays, and I have very few minor comments on this nice paper:

(i) The absence of effect on Gq and G12 (proteins is very puzzling, based on the fact that they all contain the N residue in their C-terminal region that has been proposed to be the modification site. Only G14 is somewhat affected, although its C terminal sequence is exactly the same than that of the other Gq proteins. Do the authors have an explanation?

is the N residue in Gq not modified? or is this residue modified without nevertheless affecting G protein activation?

does mutation of the asparagine residue in G14 abolished its partial inhibition by OZITX, as it does for the Gi proteins?

(ii) in their mutation experiments where the N residue supposedly modified was replaced by a alanine, the authors provide dose-response plots for only three of the subtypes (Gi3, Go and Gz) (Fig.5) and only for single dose plots for Gi1, Gi2 and GoB. Is there a reason for this?

(iii) their assay relies on the measure of Gbg release. Do the authors have an indication whether the modification impair receptor-G protein interaction or G protein activation (or trimer dissociation) ? this mzy have an impact if some processes such as precoupling are involved in the signaling process.

(iv) On top of page 9, the authors state "was again blocked in the presence of OZITX (Fig. 3D)" I presume it is Fig.4D.

Reviewer #3 (Remarks to the Author):

This is a beautiful demonstration of the use of PTX-like AB5 toxin that authors named OZITX to

selectively inhibit Gai/o and Gaz by inducing ADP-ribosylation of the Asn residue at the Ct of Ga, preventing heterotrimer and GPCR interaction. The broader and selective coverage of OZITX will be useful for many investigations, and in combination with PTX, it will allow blanket inhibition of inhibitory family G proteins. Further, similar to PTX, the authors show genetically encoded active S1 subunit of OZITX inhibits the activation. The primary assay to detect heterotrimer activation or lack thereof was detected free G $\beta\gamma$ generation by examining G $\beta\gamma$ interaction with GRK3ct. This interaction was detected using split Venus on G β and G γ and Rluc8A/nLuc on GRK2ct. Since the HEK293A Δ Gai/o CRISPR/Cas knockout cells used only carries the introduced Ga and split Venus ensures the signal is only detected from introduced G $\beta\gamma$, the signal to noise ratio is at an optimum level. I only have a few minor concerns.

First, I do not see a validation of the BRET assay for this system. While it is a robust assay, it does not necessarily demonstrate the heterotrimer dissociation but instead shows the concentration of free G $\beta\gamma$ in the system. It has been shown that upon heterotrimer activation, G $\beta\gamma$ translocate to endomembranes in a G γ -subtype dependent manner (Karunaratne et al., 2012, BBRC). While the chosen G γ here, G γ 2, has the second-highest affinity to the plasma membrane next to G γ 3, G $\beta\gamma$ 2 has exhibited a significant translocation to the endomembranes as G $\beta\gamma$ upon Gi/o-GPCR activation (Senarath et al., 2018, JBC). Since masGRK3ct-Nluc is plasma membrane associated, due to the translocation it will not capture, all the G $\beta\gamma$ generated at the plasma membrane to produce the BRET. Therefore, an assay to establish the BRET assay's validity or proper justification for not doing it is required.

Additionally, there is not sufficient discussion about endogenous G $\beta\gamma$ in HEK293 cells. These G $\beta\gamma$ should also be forming heterotrimers with the introduced Ga.

Monday, December 6, 2021

From: Dr. J. Robert Lane and Prof. Jonathan Javitch

We were pleased that all reviewers found this study of interest, and that they found OZITX will likely be a useful tool for many researchers. We would like to thank all the reviewers for their thoughtful comments and questions. We have addressed these comments as detailed below with additional data and/or discussion in the manuscript.

Sincerely,

Jonathan Javitch, MD, PhD

Rob Lane, PhD

Reviewers' comments and our response:

Reviewer #1 (Remarks to the Author): The purpose of this study is to investigate the inhibitory function of OZITX for Gi/o/z and develop experimental systems to study the signaling of Gi/o/z individually. Here are a few comments to improve the manuscript.

1. Throughout the manuscript, place a space between number and unit. For example, 10uM should be 10 uM in the Figure S2 legend.

Thank you for highlighting this error. We have made these changes throughout the revised manuscript.

2. Please provide pcDNA control (i.e. Galpha-non transfected control) for Figure S3 and Figure S4 as shown in Figure 2 or Figure 3.

These data have been added to the supplementary data as requested and are presented in Figure S3 and Figure S5

3. Please discuss a more detailed potential mechanism of non-inhibitory effects of OZITX on Gq or G12. A docking simulation of Gi/o/z and Gq or G12 binding at the active site of OZITX would show the potential mechanism.

We thank the reviewer for this suggestion. While there are crystal structures of both PTX and OZITX, there are currently no structures of any $\alpha\beta$ family toxin with a peptide substrate corresponding to the C-terminal of a G protein α subunit co-bound. Without such a structure such docking experiments are likely to be highly speculative. We have tried to gain some insight

+44 (0)115 82 30468

rob.lane@nottingham.ac.uk

into this question through other experimental approaches as detailed below in our responses to reviewers 2 and 3.

4. In p 8, lane 169, it should be Fig. 4B & C, not 3B & C.

We have corrected this mistake.

5. In p 9, lane 171, it should be Fig. 4D, not 3D.

We have corrected this mistake.

Reviewer #2 (Remarks to the Author):

The authors characterized in their paper a pertussis toxin-like protein they called OZITX that impairs activation of the G proteins from the Gi family, including Gz. Through a series of functional studies, they showed that this toxin specifically inhibited Gi/o/z activation both when preincubated or when transfected into the cells. Gz inhibition is a novel feature compared to the PTX toxin that was already used to inhibit Gi proteins. As such, this work brings a new tool that, associated to the existing ones, will undoubtedly help dissect how the Gz pathway is selectively involved in the response to Gi-coupled receptors. Taken together, this is therefore an interesting addition to the panel of tools used in GPCR signaling assays, and I have very few minor comments on this nice paper:

(i) The absence of effect on Gq and G12 proteins is very puzzling, based on the fact that they all contain the N residue in their C-terminal region that has been proposed to be the modification site. Only G14 is somewhat affected, although its C terminal sequence is exactly the same than that of the other Gq proteins. Do the authors have an explanation? Is the N residue in Gq not modified? Or is this residue modified without nevertheless affecting G protein activation? Does mutation of the asparagine residue in G14 abolished its partial inhibition by OZITX, as it does for the Gi proteins?

We thank the reviewer for picking up on this point. While the Asn residue is indeed present in $G\alpha_q$, OZITX does not inhibit G_q signalling at the level of G protein beta-gamma subunit release or intracellular Ca^{2+} mobilisation (see new data in Figure S7). In contrast, OZITX inhibits $G_{i/o/z}$ signalling at the level of G protein coupling to the receptor (see new data presented in Figure S4), G protein beta-gamma subunit release and inhibition of forskolin-stimulated cAMP. Our mutagenesis data reveals that this inhibition requires the Asn residue in the C-terminus of $G\alpha_{i/o/z}$. The lack of effect on G_q signalling can either be due to OZITX not recognising $G\alpha_q$ as a substrate and not ADP-ribosylating $G\alpha_q$ at this region or that GPCR- G_q coupling event can tolerate ADP-ribosylation of this residue.

The Asn residue is one turn down the C-terminal helix of the G protein α subunit relative to the position of the cysteine residue in $G_{\alpha_{i/o}}$ G proteins that is ADP-ribosylated by pertussis toxin. The ADP-ribosylation of this $G_{\alpha_{i/o}}$ cysteine by pertussis toxin prevents interaction of this region of the G_{α} G protein with the cytoplasmic end of TM3 of a GPCR. We would anticipate that ADP-ribosylation of the $G_{\alpha_{i/o/z}}$ asparagine residue would prevent coupling to a GPCR in a similar manner. We computationally explored the feasibility of ADP-ribosylating the asparagine of the G_{α} subunits in the $D_2R-G_{\alpha_i}$ and the $5HT_{2A}-G_{\alpha_q}$ complexes, by covalently docking the ADP-ribose moiety on the residue using the available cryo-EM structures of these complexes. Our results reveal that the $5HT_{2A}-G_{\alpha_q}$ complex can easily accommodate the ADP-ribosylated asparagine whereas this covalent docking approach could not identify any feasible pose for the $D_2R-G_{\alpha_i}$ complex without steric clash (See Author response figure 1, figure S8 in manuscript).

Author response figure 1, Figure S8. The $5HT_{2A}R-G_{\alpha_q}$ interface may accommodate ADP-ribosylation of the C terminal asparagine of G_{α_q} . In the cryo-EM structure of $D_2R-G_{\alpha_i}$ complex (PDB 7JVR)⁴⁹, Asn³⁴⁷, eight residues from the C-terminus of G_{α_i} , is in close proximity to intracellular loop 2 (IL2) of D_2R (A), and is tightly packed with Tyr¹⁴² and Asn¹⁴³ of IL2 (B), which prevented ADP-ribosylation of the residue in our covalent docking study. In contrast, in the $5HT_{2A}R-G_{\alpha_q}$ complex (PDB 6WHA)⁴⁸, the aligned Asn²³⁹ in the engineered G_{α_q} is far from IL2 of $5HT_{2A}R$ and faces a more open space (C), which allows easy in silico ADP-ribosylation of Asn²³⁹. Multiple possible poses of the covalently attached ADP-ribose moiety can be generated in our docking study, a representative pose is shown in panel D. The covalent docking was carried out with the covDock module of Schrodinger suite (version 2021-1), assuming a SN2 nucleophilic substitution reaction, which results in the α -orientation of the attached ADP-

ribose moiety on the asparagine⁵². It should be noted that the 5HT_{2A}R structure is in complex with an engineered Gα_q protein in which the N-terminal 35 residues are replaced by the corresponding Gα_{i2} residues⁴⁸, which are only minimally involved in the receptor-G protein interface.

However, from these studies we cannot discount the possibility that an ADP-ribosylated Gα_q would be unable to couple to its cognate GPCR and we should note that the 5HT_{2A}R structure is in complex with an engineered Gα_q protein in which the N-terminal 35 residues are replaced by the corresponding Gα_{i2} residues⁴⁸, which are only minimally involved in the receptor-G protein interface. (See Author response figure 1, Figure S8 in manuscript). We extended our analysis of OZITX's action to 5HT_{2A}R-Gq signalling (see Author Response Figure 2). Our data reveal that OZITX is unable to inhibit M₁ mAChR, M₃ mAChR (Ca²⁺ data), NTS₁R, and 5HT_{2A}R coupling to Gα_q, suggesting that this lack of inhibition is not specific to different GPCR-Gα_q receptor pairs. Note that in the case of 5HT_{2A}R we observe an increase in maximal effect of 5HT_{2A} in the presence of both S1-PTX and S1-OZITX expression. This may reflect inhibition of Gi/o signalling from activation of endogenously expressed 5HT_{1D}R in HEK293 cells (see Atwood, B.K., Lopez, J., Wager-Miller, J. *et al. BMC Genomics* **12**, 14 (2011). <https://doi.org/10.1186/1471-2164-12-14>) that releases beta-gamma-venus subunits for our measurements of Gα_q activation.

5HT_{2A} /Gα_q βγ release assay

Author Response Figure 2: Cells were transfected with the 5HT_{2A}R, Gα_q, the G protein activation sensors and either a pcDNA3.1+ control (teal triangles), OZITX-S1 cDNA (black circles) or PTX-S1 cDNA (pink squares). Cells were then stimulated with increasing concentrations of 5-HT before BRET detection. Data represents the mean drug induced increase in BRET ratio from vehicle ± SEM.

Previous studies have shown that replacement of the 5 C-terminal amino acids of $G\alpha_q$ with that of $G\alpha_i$ allows this chimeric G protein to couple to $G\alpha_{i/o}$ coupled receptors. Importantly, however, these 5 residues, which include the cysteine that is modified by PTX, are not sufficient to confer sensitivity to PTX as chimeric $G\alpha_q$ mutants harbouring the last five carboxy-terminal residues of $G\alpha_{i2}$ or $G\alpha_o$ are resistant to pertussis toxin-catalyzed ADP-ribosylation (*FEBS letters* **441**, 67-70 (1998)). As we discussed in the original submission of this manuscript, this suggests that although the N-terminal Cys is the substrate, other regions of the G protein likely contribute to the selective action of PTX at $G\alpha_{i/o}$. In agreement with this idea, a peptide based on the last 20 amino acids of $G\alpha_{i3}$ were found to be a substrate of PTX but shortening the length of this peptide to 10 amino acids caused a decrease in both K_m and V_{max} . We hypothesized that OZITX's selective action at $G\alpha_{i/o/z}$ G proteins over the Gq family G proteins even though the target Asn that is ADP-ribosylated is present in the Gq family may be because this selectivity is driven in part by other regions of the G protein.

We generated $G\alpha_{qi10}$ and $G\alpha_{qi13}$ constructs that can couple to $G\alpha_{i/o/z}$ coupled receptors such as the D_2R but act to initiate a Gq signalling cascade to increase intracellular Ca^{2+} levels. Unlike $G\alpha_{qi5}$ the coupling of the $G\alpha_{qi10}$ chimera and the $G\alpha_{qi13}$ chimera were inhibited by PTX and meaning that the last 10 amino acids are sufficient to confer PTX sensitivity. OZITX also inhibited the Ca^{2+} signal from both G α G protein chimeras. While both PTX and OZITX partially inhibited the $G\alpha_{qi10}$ Ca^{2+} signal, expression of both toxins completely inhibited $G\alpha_{qi13}$ signalling. This pattern is consistent with the idea that the greater the amount of C-terminal $G\alpha_i$ amino acids swapped with those of $G\alpha_q$, the chimeric $G\alpha_{qi}$ proteins become a better substrate for both $\alpha\beta$ toxins. However, the lower potency of ropinirole in the $G\alpha_{qi13}$ Ca^{2+} assay suggests that the coupling of the D_2R to this chimera is less efficient which may account for the apparently greater inhibitory effect of PTX and OZITX. Thus, although these new data are consistent with $G\alpha_{i/o/z}$ G proteins being a substrate for OZITX whereas $G\alpha_q$ family G proteins are not, we cannot discount the possibility that Gq family G proteins are ADP-ribosylated by OZITX and but that this modification has no impact on Gq signalling. These data have been added to the manuscript and discussed in the results section under the heading 'The C-terminal 10 residues of Gai are sufficient to confer OZITX selectivity'. These data along with our docking study is discussed further in the Discussion section, page 15 paragraph 1. Further studies are required to definitively understand the basis of OZITX selectivity but are beyond the scope of this manuscript which describes this selectivity and identifies mutant G protein alpha subunits that are resistant to its effect.

IN PARTNERSHIP:

The Universities of Birmingham and Nottingham

Author Response Figure 2, Figure S7 in revised manuscript: The C-terminal 10 amino acids of Gα_i are sufficient to confer OZITX selectivity. HEK293 cells were transfected with plasmids encoding the M₃ mAChR and Gα_q (A & B) or the D_{2L}R and Gα_q (C & D) or chimeras Gα_qi10 (E & F, in which last 10 amino acids of Gα_q were replaced with those of Gα_i) and Gα_qi13 (G & H, in which last 13 amino acids of Gα_q were replaced with those of Gα_i and the catalytic subunit of PTX or OZITX). The Ca²⁺ mobilisation response to 30 μM carbachol (A) or ropinirole (C, E, G) was measured over 180 seconds. The area under the curve (AUC) of Ca²⁺ mobilisation responses over 180 seconds to increasing concentrations of carbachol (B) or ropinirole (D,F,H) were measured.

IN PARTNERSHIP:

The Universities of Birmingham and Nottingham

(ii) in their mutation experiments where the N residue supposedly modified was replaced by an alanine, the authors provide dose-response plots for only three of the subtypes (Gi3, Go and Gz) (Fig.5) and only for single dose plots for Gi1, Gi2 and GoB. Is there a reason for this?

We appreciate the reviewer's comment. We have now added these data to the supplementary data as figure S6.

(iii) their assay relies on the measure of Gbg release. Do the authors have an indication whether the modification impairs receptor-G protein interaction or G protein activation (or trimer dissociation)? This may have an impact if some processes such as precoupling are involved in the signaling process.

The reviewer raises an important point. We have now added additional data in which we measure the recruitment of $G\alpha_{oA}:G\beta:G\gamma$ -venus or $G\alpha_z:G\beta:G\gamma$ -venus G protein heterotrimer to the $D_{2L}R$ in the presence of either PTX-S1 or OZITx-S1 expression. Our data clearly show that PTX inhibits $G\alpha_{oA}:G\beta:G\gamma$ -venus G protein heterotrimer recruitment but not $G\alpha_z:G\beta:G\gamma$ -venus to the D_2R as expected. OZITx inhibits both $G\alpha_{oA}:G\beta:G\gamma$ -venus and $G\alpha_z:G\beta:G\gamma$ -venus G protein heterotrimer recruitment to the D_2R . This is consistent with a mechanism of action whereby the ADP-ribosylation of the C-terminal asparagine of $G\alpha_{i/o/z}$ by OZITx prevents their coupling to GPCRs. These data have been added to the manuscript as figure S4 and discussed on page 9, line 24.

Figure S4: $G\alpha_{oA}:\beta_1:\gamma_2$ -venus (A) or $G\alpha_z:\beta_1:\gamma_2$ -venus (B) recruitment to the $D_{2L}R$ fused to nanoluciferase ($D_{2L}R$ -NLuc) at its C-terminus. HEK293 cells were transiently transfected with plasmids encoding $D_{2L}R$ -NLuc, $G\alpha_{oA}$ or $G\alpha_z$, $G\beta_1$ and $G\gamma_2$ -venus and either the catalytic subunit of PTX (S1-PTX – pink squares), OZITx (S1-OZITx, black circles) or pcDNA3 as a control (teal triangles). The ability of increasing concentrations of the D_2R agonist ropinirole to cause recruitment of the $G\alpha_{oA}:\beta:\gamma$ -venus or $G\alpha_z:\beta:\gamma$ -venus heterotrimer to D_{2L} -NLuc was measured as an increase in BRET ratio.

(iv) On top of page 9, the authors state “was again blocked in the presence of OZITX (Fig. 3D)” I presume it is Fig.4D.

This error has been corrected. Thanks for pointing this out.

Reviewer #3 (Remarks to the Author):

This is a beautiful demonstration of the use of PTX-like AB5 toxin that authors named OZITX to selectively inhibit $G_{\alpha i/o}$ and $G_{\alpha z}$ by inducing ADP-ribosylation of the Asn residue at the C-terminus of G_{α} , preventing heterotrimer and GPCR interaction. The broader and selective coverage of OZITX will be useful for many investigations, and in combination with PTX, it will allow blanket inhibition of inhibitory family G proteins. Further, similar to PTX, the authors show genetically encoded active S1 subunit of OZITX inhibits the activation. The primary assay to detect heterotrimer activation or lack thereof was detected free $G\beta\gamma$ generation by examining $G\beta\gamma$ interaction with GRK3ct. This interaction was detected using split Venus on $G\beta$ and $G\gamma$ and Rluc8A/nLuc on GRK2ct. Since the HEK293A $\Delta G_{\alpha i/o}$ CRISPR/Cas knockout cells used only carries the introduced G_{α} and split Venus ensures the signal is only detected from introduced $G\beta\gamma$, the signal to noise ratio is at an optimum level. I only have a few minor concerns.

First, I do not see a validation of the BRET assay for this system. While it is a robust assay, it does not necessarily demonstrate the heterotrimer dissociation but instead shows the concentration of free $G\beta\gamma$ in the system. It has been shown that upon heterotrimer activation, $G\beta\gamma$ translocate to endomembranes in a $G\gamma$ -subtype dependent manner (Karunarathne et al., 2012, BBRC). While the chosen $G\gamma$ here, $G\gamma 2$, has the second-highest affinity to the plasma membrane next to $G\gamma 3$, $G\beta\gamma 2$ has exhibited a significant translocation to the endomembranes as $G\beta\gamma$ upon $G_{i/o}$ -GPCR activation (Senarath et al., 2018, JBC). Since masGRK3ct-Nluc is plasma membrane associated, due to the translocation it will not capture, all the $G\beta\gamma$ generated at the plasma membrane to produce the BRET. Therefore, an assay to establish the BRET assay's validity or proper justification for not doing it is required. Additionally, there is not sufficient discussion about endogenous $G\beta\gamma$ in HEK293 cells. These $G\beta\gamma$ should also be forming heterotrimers with the introduced G_{α} .

We thank the reviewer for this comment. This assay has been extensively characterized in previous studies (see Masuho I, Martemyanov KA, Lambert NA. Monitoring G Protein Activation in Cells with BRET. Methods Mol Biol. 2015;1335:107-13. doi: 10.1007/978-1-4939-2914-6_8. PMID: 26260597; PMCID: PMC4879496 & Masuho I, Ostrovskaya O, Kramer GM, Jones CD, Xie K, Martemyanov KA. Distinct profiles of functional discrimination among G proteins determine the actions of G protein-coupled receptors. Sci Signal. 2015 Dec 1;8(405):ra123. doi: 10.1126/scisignal.aab4068. PMID: 26628681; PMCID: PMC4886239). The reviewer is correct that what the assay detects is changes in free \$\beta\gamma\$ -venus subunits (for example following stimulation by a particular agonist). Of note however, without the over-expression of the \$G_{\alpha}\$

IN PARTNERSHIP:

The Universities of Birmingham and Nottingham

subunit we observe only small change in BRET as compared to when we overexpress the $G\alpha$ subunit (see pcDNA control all experiments using these sensors). Overexpression of the $G\alpha$ subunit decreases the basal BRET relative to the pcDNA control consistent with this overexpression sequestering $\beta\gamma$ -venus subunits away from the mas-GRK-Nluc. Thus, in our experimental design we consider an increase in BRET to be a measurement of heterotrimeric G protein activation, and as shown by our data it allows us to look at specific receptor-G protein subtype coupling events. We acknowledge we cannot use this assay to measure translocation of $G\beta\gamma$ to endomembranes. However, in terms of characterizing the action of the toxin we have shown it inhibits $G\alpha_{i/o/z}$ signalling at the level of GPCR-G protein coupling (see new data in which we measure interaction between the D2R and $G\alpha_{i/o/z} + \beta + \gamma$ -venus in response to reviewer 2, Figure S4), at the level of $\beta\gamma$ -venus release and at the level of a downstream second messenger (inhibition of cAMP). While we have also used the $\beta\gamma$ -venus release assay to show that OZITX is ineffective at inhibiting $G\alpha_q$, $G\alpha_{12/13}$ and $G\alpha_s$ family G proteins, we have confirmed the lack of effect at $G\alpha_q$ in a Ca^{2+} assay (see new figure S7). In summary, we feel that even though we are not specifically measuring translocation of $G\beta\gamma$ to endomembranes we have provided additional evidence of OZITX's pattern of selectivity and mode of action to prevent functional GPCR- $G\alpha_{i/o/z}$ coupling such that this does not impact the conclusions of our study.

IN PARTNERSHIP:

The Universities of Birmingham and Nottingham

REVIEWERS' COMMENTS:

Reviewer #1 (Remarks to the Author):

The revised manuscript addressed all the concerns that I have raised.

Reviewer #2 (Remarks to the Author):

The authors appropriately addressed all my concerns. I have no additional issue for this nice work to be published.

Reviewer #3 (Remarks to the Author):

The authors have sufficiently addressed my concerns.